**Matters arising**

# Reply to: Re-examining extreme carbon isotope fractionation in the coccolithophore *Ochrosphaera neapolitana*

Yi-Wei Liu [1] ✉, Robert A. Eagle[2,3] ✉, Sarah M. Aciego[4], Rosaleen E. Gilmore[3] & Justin B. Ries[5]

REPLYING TO H. Zhang et al. *Nature Communications* https://doi.org/10.1038/s41467-022-35109-4 (2022)

In the accompanying Matters Arising, Zhang et al.[1] report that they did not generate the same fractionation in $\delta^{13}$C between seawater dissolved inorganic carbon (DIC) and both inorganic and organic carbon of the coccolithophore *Ochrosphaera neapolitana* that were observed in our study[2]. These differences in findings are an opportunity to discuss the uncertainties and limitations of different designs of ocean acidification experiments. Here, we note that Zhang et al.[1] did not closely replicate our experimental conditions, meaning that their alternative hypotheses about what could be occurring in our experiment have ultimately not been tested. Additionally, we highlight elements of their analyses that are under-constrained and how it is difficult to fully resolve these issues without more experimental work.

## Clarification on nominal $p$CO$_2$ levels

To clarify, $p$CO$_2$ labels assigned to the treatments in our study were only estimates based upon relative flow rates of the different gases through uncalibrated mass flow controllers, and not the measured $p$CO$_2$ of the mixed gases. This is why we refer to those labels as "nominal" $p$CO$_2$ treatments and also present the actual $p$CO$_2$ values, calculated from measured total alkalinity (TA) and DIC.

## Clarification on the method, timescales, and theoretical analyses

In our original study[2], we clearly state our assumptions in modeling the $\delta^{13}$C of DIC in the experiment and provide experimental data on the $\delta^{13}$C of DIC from a similar experiment using the same reagents. The theoretical analyses presented by Zhang et al.[1] on timescales of equilibration in hypothetical experiments don't invalidate our assumptions because they show that we pre-bubbled our experimental treatments long enough for $p$CO$_2$ of the air and mixed gases to achieve equilibrium.

To further clarify this matter, the treatments in our experiment were continuously bubbled at the rate of -1.5 L/min with the mixed gases using a microporous bubbler for 2 weeks before the experiment and throughout its 2-week duration. Owing to this rapid bubbling rate (-1.5 L/min), the blue curves (corresponding to "slow-bubbling" rates) in Figure 2 and Supplementary Figure 2 of Zhang et al.[1] are not applicable to our experiment. Moreover, under the "fast-bubbling" scenario (red curves in Figure 2 and Supplementary Figure 2 of ref. 1.), isotopic equilibrium was achieved even for the lowest $p$CO$_2$ treatment by day 14 of pre-bubbling (with -66% equilibrium achieved by day 5). Thus, according to the model of Zhang et al.[1], it seems reasonable to assume that $p$CO$_2$ and $\delta^{13}$C of the mixed gases and seawater treatments should have been close to equilibrium throughout the duration of the experiment.

Zhang et al.[1] estimate a potential exchange rate ($k_E$) for our study. However, one cannot assess the accuracy of this estimate without knowing more about the experimental and model systems from which this estimate was derived (for example, the distributions of gas bubble sizes), which are not available. However, we acknowledge that we should have more clearly described the duration and rates of bubbling in the methods section of the original manuscript.

## Differences in experimental design and methodology

In our original study[2], we acknowledged the potential limitations to our approach and the assumptions involved. We explicitly acknowledged that we estimated seawater $\delta^{13}$C$_{DIC}$ (used in the calculation of particulate organic carbon (POC) $\Delta^{13}$C and particulate inorganic carbon (PIC) $\Delta^{13}$C) from seawater solutions that were formulated similarly to those used in the culture experiment, as original culture waters were not preserved for $\delta^{13}$C$_{DIC}$ measurements. Whilst the experimental data would have been better constrained by direct measurements of culture media during the course of the experiment, we also note that the experimental data presented by Zhang et al.[1] is under-constrained, as

[1]Institute of Earth Sciences, Academia Sinica, 128, Sec. 2, Academia Road, Nangang, Taipei 11529, Taiwan. [2]Institute of the Environment and Sustainability, University of California—Los Angeles, La Kretz Hall, 619 Charles E. Young Dr. E #300, Los Angeles, CA 90024, USA. [3]Atmospheric and Oceanic Sciences Department, University of California—Los Angeles, Math Sciences Building, 520 Portola Plaza, Los Angeles, CA 90095, USA. [4]Department of Geology and Geophysics, University of Wyoming, 1000 East University Avenue, Laramie, WY 82071-2000, USA. [5]Department of Marine and Environmental Sciences, Marine Science Center, Northeastern University, 430 Nahant Road, Nahant, MA 01908, USA. ✉e-mail: liuyiwei@earth.sinica.edu.tw; robeagle@g.ucla.edu

discussed below, which might contribute to differences in experimental outcomes between the two studies.

First, the basic carbonate chemistries of the two experiments were different, even for what Zhang et al.[1] assert are comparable $pCO_2$ treatments. The DIC and $[CO_{2aq}]$ in the beginning of the Zhang et al.[1] experiment was 2050 and ~4.2 μmol/kg-SW, respectively, while ours were ~2400 and ~6.4 μmol/kg-SW, yielding ~50% differences in some carbonate system parameters. Previous work has shown that coccolithophores are sensitive to the concentration of dissolved $CO_2$, as well as other carbonate system parameters, in seawater[3,4].

Second, Zhang et al.[1] manipulated $pCO_2$ and carbonate chemistry of their single treatment by adding $NaHCO_3$ and HCl and then sealing the culture vessels, rather than continuously bubbling the treatment with a mixed gas formulated at the desired $pCO_2$ condition as done in our study[2]. The purpose of the continuous bubbling approach used in our original study[2] is to maintain relatively stable carbonate chemistry throughout the experiment. In contrast, the carbonate chemistry in the Zhang et al.[1] experiment would have changed substantially as the coccolithophores removed $CO_2$ through photosynthesis (reducing DIC) and $CO_3^{2-}$ through calcification (reducing TA) during their exponential growth throughout the experiment, yielding a chemically unconstrained "drift experiment".

Third, Zhang et al.[1] modeled a +3‰ shift of $\delta^{13}C$ when DIC was consumed for photosynthesis in a closed system with a cell density of $10^5$ cell/mL. The continuous bubbling approach employed in our experiment would also minimize the photosynthesis-driven Rayleigh fractionation of $\delta^{13}C$ of DIC during the experiment because the carbon pool was continuously replenished with $CO_2$ of a constant isotopic composition throughout the experiment. Therefore, the DIC available for calcification should be nearly identical throughout the duration of the experiment, rather than evolving with increasing cell density as described in Figure 3 of Zhang et al.[1]. Furthermore, the cell densities in our treatments were extremely low, with the "100,000 cell/mL" reported as a conservative upper bound.

Zhang et al.[1] also state that they only measured pH and DIC, from which they calculated $pCO_2$, at the end of their drift experiment. Thus, the $pCO_2$ that they report and compare to ours is, in fact, not what the coccolithophores experienced while they were actually recording their isotopic signatures. These conditions would be expected to be quite different from the end-point conditions that were reported given that the culture vessels were sealed, not continuously bubbled with gases formulated at the appropriate $pCO_2$, contained a small volume of seawater (150 mL), and hosted a rapidly growing population of photosynthesizing and calcifying algae. That Zhang et al.[1] reported a PIC:POC ratio for the coccolithophores of only 0.1:1 indicates that DIC would have been drawn down faster than TA, causing substantial declines in $pCO_2$ throughout the experiment that were neither quantified nor reported by the authors. Based upon the initial DIC that they report (2050 μmol/kg-SW), their reported cell density, and PIC and POC content per cell, it can be estimated that $pCO_2$ would have decreased by 300 to 400 ppm throughout their experiment—although this estimation is complicated by the lack of measured carbonate system parameters (beyond initial DIC) reported for their experiment.

Furthermore, Zhang et al.[1] measured the end-point of their drift experiment with a liquid junction glass pH electrode calibrated with NBS (freshwater) pH buffers, rather than seawater pH buffers. Calibrating a liquid junction glass pH electrode with NBS buffers and then measuring a seawater solution imparts up to 0.2 error in measured pH because of the well-known liquid junction potential bias[5]. This error is not corrected for simply by recalculating pH on a different scale because it is a matrix effect that arises from the difference in salinities between the calibration solutions and the measured samples. This would impart an additional error in the calculated $pCO_2$ of 200–300 ppm for the range of carbonate system parameters estimated from the sparse measured data provided by Zhang et al.[1].

Finally, Zhang et al.[1] calculated $pCO_2$ of their experimental treatment from pH and DIC. However, it is well-established that carbonate chemistry of $CO_2$-manipulation studies should be calculated from measured TA and DIC[6] because measurement of pH using a liquid junction glass pH electrode (as done by Zhang et al.[1]) has significantly lower analytical resolution (2 significant figures, and less if calibrated with NBS buffers) than measurement of TA by Gran titration (3–4 significant figures). This can impart additional error of 100–200 ppm in calculated $pCO_2$.

Collectively, these uncertainties in calculated $pCO_2$ in the experiment by Zhang et al.[1] could amount to several 100s of ppm, even up to 1000 ppm. These potential sources of uncertainties were not explored or well acknowledged in their comment.

It is also worth noting differences in the source of the two isolates of *O. neapolitana* (RCC1357 versus an isolate derived from Bigelow Laboratory for Ocean Sciences) used in these studies, which prior work[7] has shown can yield phenotypic differences that impart different degrees of mass-dependent isotopic fractionation.

## Conclusion

It is difficult to accurately assess the total uncertainty arising from the experiment by Zhang et al.[1] due to the scarcity of reported carbonate system parameters, unconstrained drift in carbonate and isotopic chemistry owing to drawdown in DIC and TA by photosynthesis and calcification within their closed experimental system, error in pH measurement from liquid junction bias, uncertainty associated with calculation of $pCO_2$ from pH and DIC (rather than from TA and DIC), and the different sources of the *O. neapolitana* isolates. Given the extent of these differences in experimental design and methodology, we do not believe that the two sets of experimental data have been proven to be in conflict. The actual $pCO_2$ experienced by *O. neapolitana* during their period of growth in Zhang et al.[1] might have been quite different from the values that they report, and it cannot be ruled out that corrected values would fall on the trajectory of the $\Delta^{13}C$ versus $pCO_2$ trends observed for PIC and POC in our experiment (Figure 4 in ref. [2], Figure 1 in ref. [1]). This exchange highlights important issues with both experimental approaches that are difficult to reconcile in this short-format response, but would certainly be worth considering and investigating in future experiments on the response of calcifying phytoplankton to $CO_2$-induced ocean acidification.

## Data availability

All results of the culture experiment and isotopic analyses are presented in the original manuscript and Supplementary Table 1 of ref. [2]. The data is also available upon request from the corresponding authors.

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

## Author contributions

Y.-W.L., R.A.E., S.M.A., R.E.G., and J.B.R. participated in the discussion and writing of this article.

## Competing interests

The authors declare no competing interests.
