## [Peer Review File · Nature Communications]

Reply to: No extreme carbon isotope fractionation in the coccolithophore *Ochrosphaera neapolitana*Referees' comments:

Reviewer #1 (Remarks to the Author):

Following the recent appearance of the paper "A coastal coccolithophore maintains pH homeostasis and switches carbon sources in response to ocean acidification" by Lui et al., Zhang et al. have submitted a comment entitled «No extreme carbon isotope fractionation in the coccolithophore *Ochrosphaera neapolitana*. Here, I expose my opinion on both the comment and the reply.

The main source of disagreement relates on the methods and in particular the isotopic composition of the medium against which the composition of the biomineral and organic matter of the microalgae must be reported.

My personal opinion is that the values C-isotope fractionation originally presented by Liu et al. are indeed not realistic and potentially biased by the design of the experiment. I had this assessment when the paper first came out. The lack of documentation of the isotopic composition of the medium from which *O. neapolitana* grew is a serious omission (initial and final state). I am somehow surprised that the Reviewers of the paper did not spot this. This is all the more regrettable that, yet in my view, the take-home message of the paper is not guaranteed by a solid and reliable dataset.

On this respect I think that the publication of this Matter Arising seems justified. Perhaps, I can recommend that exploring the inorganic-organic C isotope offset can be a clue for constraining the isotopic composition of the medium, as you have two equations for only one unknown under the premise that calcite and organic matter formed simultaneously from the same fluid. Just an idea, I'm happy to be wrong.

In their reply, Liu and colleagues are right in stating the results presented by Zhang and al. (coming from a set of experiment that the latter have undertaken for the purpose of this discussion) are not directly comparable to the original dataset. In the bubbling method was applied to perturb the carbonate chemistry of the method, while in the commentary, the alkalinity method was. This prevents the direct comparison as you are changing other parameters than $[\text{CO}_2 \text{ aq}]$ and pH. This is a side note but I fully endorse the comment made by Liu et al. in their reply about the pH measurement and the need to calibrate the device with a Tris buffer solution. It is also a good practice to express the pH values on the Total Scale.

Reviewer #2 (Remarks to the Author):

I have to say that I did not find the reply by Zhang et al. convincing. First of all, the problem of a potential isotopic disequilibrium is unconnected to the difference between the 'nominal' targeted $p\text{CO}_2$ and actually calculated levels (from TA and DIC). Even if both would have been the same, there is the possibility of isotopic disequilibrium as chemical equilibration is significantly quicker. How quickly it is reached depends on a number of parameters including aeration bubble size, water volume and flow rate. But simple notions of the actual flow rate (1.5 L/min), the duration (2 weeks) or terms such as 'microporous' or 'vigorously' do not help. In fact, without measured evidence of isotopic equilibrium the possibility of non-equilibration can not be ruled out and hence might explain the observed differences pointed out in Figure 1 of Zhang et al.

In their reply to Zhang et al. Liu et al. make further five points which I am going to address separately below.

1) Indeed, the carbonate chemistry speciation in Liu et al. is quite unique in comparison to other studies and typical seawater conditions. As shown in their supplementary Figure 2, total alkalinity concentrations are almost 1000 $\mu\text{mol/kg}$ higher (i.e. 30%) than for typical seawater with a salinity of 36. This leads to significantly higher HCO_3^- and CO_3^{2-} concentrations for the same pH/ $p\text{CO}_2$ than in other studies. However, higher dissolved inorganic carbon availability in comparison to

other studies should have lead to higher isotopic fractionation, not lower, as of increased supply over demand, contrary to what Liu et al. did find.

On this note, the artificially elevated TA to salinity ratio in Liu et al. is most likely the result of using the instant ocean salt mixture. It is indicative of a non-standard ion matrix, which makes calculating pCO₂, pH and other carbonate chemistry parameters from measured DIC and TA for a certain salinity problematic at best, i.e. it is prone to significant bias in calculated pCO₂/pH (see also comment below).

2) Contrary to the claim by Liu et al. continuous bubbling rather than following a dilute-batch closed-system approach is not a more 'recommended' or correct approach. Both have advantages and disadvantages one has to be aware of. However, they equally require that carbonate chemistry speciation is actually measured, as is d13C_DIC, to account for DIC and TA changes during photosynthesis. On this note, while bubbling is replenishing DIC (at least to some degree), TA is still being consumed, hence carbonate chemistry speciation is shifting during coccolithophore growth as well in the Liu et al. approach. Again, that is not a problem, but it has to be measured and reported.

3) I agree, that using the end-point DIC/TA is not representative for the incubation pH/pCO₂, and typically a mean for the start and end conditions is taken, but this fact does not help explaining the measured isotopic off-set shown in Figure 1 of Zhang et al.

4) Again, I agree that using a NBS calibrated pH electrode to calculate carbonate chemistry speciation is far from ideal and there can be a substantial bias, but this would simply shift the data of Zhang et al. shown in Figure 1 horizontally, while it would need vertical shifting (unless the pCO₂ would have been underestimated in an extreme fashion - see below).

5) As already pointed out in comment #4, potentiometric pH measurements with a NBS-calibrated electrode is not best practice, but can not explain the general isotopic offset between the various data sets. However, for the one data point created by Zhang et al., there is the possibility that pCO₂ was underestimated as of non-ideal NBS-based potentiometric pH measurements (although the magnitude estimated by Liu et al. is most likely extreme). If indeed the case, this would suggest that *Ochrosphaera* could be different to all other coccolithophores tested before. Given the potential problems with the instant ocean seawater and carbonate chemistry calculations mentioned above, and the fact that the data set by Liu et al. appears to stick out rather than the data point generated by Zhang et al., I would rather tend to go with the simplest explanation for the general discrepancy shown in Figure 1, i.e. that isotopic equilibrium was not reached in the original Liu et al. data set. However, a final verdict would require additional data.

Reviewer #1 (Remarks to the Author):

Following the recent appearance of the paper “A coastal coccolithophore maintains pH homeostasis and switches carbon sources in response to ocean acidification” by Lui et al., Zhang et al. have submitted a comment entitled «No extreme carbon isotope fractionation in the coccolithophore *Ochrosphaera neapolitana*. Here, I expose my opinion on both the comment and the reply.

The main source of disagreement relates on the methods and in particular the isotopic composition of the medium against which the composition of the biomineral and organic matter of the microalgae must be reported.

My personal opinion is that the values C-isotope fractionation originally presented by Liu et al. are indeed not realistic and potentially biased by the design of the experiment. I had this assessment when the paper first came out. The lack of documentation of the isotopic composition of the medium from which *O. neapolitana* grew is a serious omission (initial and final state). I am somehow surprised that the Reviewers of the paper did not spot this. This is all the more regrettable that, yet in my view, the take-home message of the paper is not guaranteed by a solid and reliable dataset.

The lack of data for the isotopic composition of the growing medium was not an ‘omission’. We acknowledged in the original manuscript that these data were estimated using the data that were available.

On this respect I think that the publication of this Matter Arising seems justified. Perhaps, I can recommend that exploring the inorganic-organic C isotope offset can be a clue for constraining the isotopic composition of the medium, as you have two equations for only one unknown under the premise that calcite and organic matter formed simultaneously from the same fluid. Just an idea, I’m happy to be wrong.

Interesting idea but we believe this would require the assumption that the mass dependent fractionation between the water and tissue and water and shell is not impacted by pCO₂ treatment, which it may be, rendering the set of equations underconstrained.

In their reply, Liu and colleagues are right in stating the results presented by Zhang and al. (coming from a set of experiment that the latter have undertaken for the purpose of this discussion) are not directly comparable to the original dataset. In the bubbling method was applied to perturb the carbonate chemistry of the method, while in the commentary, the alkalinity method was. This prevents the direct comparison as you are changing other parameters than [CO₂ aq] and pH. This is a side note but I fully endorse the comment made by Liu et al. in their reply about the pH measurement and the need to calibrate the device with a Tris buffer solution. It is also a good practice to express the pH values on the Total Scale.

We agree that the fundamental differences in the design of these two experiments renders their results incomparable.

Reviewer #2 (Remarks to the Author):

I have to say that I did not find the reply by Zhang et al. convincing. First of all, the problem of a potential isotopic disequilibrium is unconnected to the difference between the 'nominal' targeted pCO₂ and actually calculated levels (from TA and DIC). Even if both would have been the same, there is the

possibility of isotopic disequilibrium as chemical equilibration is significantly quicker. How quickly it is reached depends on a number of parameters including aeration bubble size, water volume and flow rate. But simple notions of the actual flow rate (1.5 L/min), the duration (2 weeks) or terms such as 'microporous' or 'vigorously' do not help. In fact, without measured evidence of isotopic equilibrium the possibility of non-equilibration can not be ruled out and hence might explain the observed differences pointed out in Figure 1 of Zhang et al.

We agree that Zhang's focus on the offset between target and actual $p\text{CO}_2$ is not relevant because isotopic disequilibrium could still exist even if the gas and solution are in equilibrium with respect to $p\text{CO}_2$. However, we responded to this point in our reply because Zhang et al argue that if the gas $p\text{CO}_2$ is different than the seawater $p\text{CO}_2$ —which we show is not the case—then the air and gas cannot be at equilibrium. This does not necessarily mean that gas and seawater are in isotopic equilibrium (as isotopic equilibration takes longer), but it does counter their argument that the apparent offset between gas and seawater $p\text{CO}_2$ is evidence of isotopic disequilibrium.

In their reply to Zhang et al. Liu et al. make further five points which I am going to address separately below.

Indeed, the carbonate chemistry speciation in Liu et al. is quite unique in comparison to other studies and typical seawater conditions. As shown in their supplementary Figure 2, total alkalinity concentrations are almost 1000 $\mu\text{mol}/\text{kg}$ higher (i.e. 30%) than for typical seawater with a salinity of 36. This leads to significantly higher HCO_3^- and CO_3^{2-} concentrations for the same pH/ $p\text{CO}_2$ than in other studies. However, higher dissolved inorganic carbon availability in comparison to other studies should have led to higher isotopic fractionation, not lower, as of increased supply over demand, contrary to what Liu et al. did find.

We understand this point, but the increased fractionation expected from increased supply of substrate relative to demand assumes that demand for DIC for photosynthesis by coccolithophores will not increase with $p\text{CO}_2$ and DIC— which is a tenuous assumption. It is indeed possible that supply and demand for substrate would increase in lock-step up to the point that DIC is no longer limiting for coccolithophore photosynthesis.

On this note, the artificially elevated TA to salinity ratio in Liu et al. is most likely the result of using the instant ocean salt mixture. It is indicative of a non-standard ion matrix, which makes calculating $p\text{CO}_2$, pH and other carbonate chemistry parameters from measured DIC and TA for a certain salinity problematic at best, i.e. it is prone to significant bias in calculated $p\text{CO}_2/\text{pH}$ (see also comment below).

The ion matrix for type of instant ocean used in this study is actually very similar to that of seawater and has been used in many ocean acidification experiments where carbonate system parameters are calculated from DIC and TA. The main difference is the slightly elevated total alkalinity—but even that is comparable to many tropical carbonate platform systems in nature. There are other forms of Instant Ocean that are much more different from seawater, but those were not used.

2) Contrary to the claim by Liu et al. continuous bubbling rather than following a dilute-batch closed-system approach is not a more 'recommended' or correct approach. Both have advantages and disadvantages one has to be aware of. However, they equally require that carbonate chemistry speciation is actually measured, as is $\delta^{13}\text{C}_{\text{DIC}}$, to account for DIC and TA changes during

photosynthesis. On this note, while bubbling is replenishing DIC (at least to some degree), TA is still being consumed, hence carbonate chemistry speciation is shifting during coccolithophore growth as well in the Liu et al. approach. Again, that is not a problem, but it has to be measured and reported.

We did measure and report TA and DIC in our experiment. Zhang et al did not. They measured pH and DIC, and their pH measurements were made incorrectly. We also disagree that continuous bubbling is not the preferred approach for OA experiments. The reason is that the ultimate goal of these experiments is to maintain a constant $p\text{CO}_2$. Constant bubbling of a mixed gas of fixed $p\text{CO}_2$ is the best way to ensure a near constant $p\text{CO}_2$, even if TA is drifting down due to calcification. In our system, $p\text{CO}_2$ stays constant, while TA drifts downward, which brings DIC downward slightly. In the Zhang et al approach, all three parameters ($p\text{CO}_2$, DIC, and TA) drift downward, and none of these were actually measured by them, except for DIC prior to the start of their experiment.

3) I agree, that using the end-point DIC/TA is not representative for the incubation pH/ $p\text{CO}_2$, and typically a mean for the start and end conditions is taken, but this fact does not help explaining the measured isotopic off-set shown in Figure 1 of Zhang et al.

What it shows is that the carbonate chemical system within the Zhang et al experiment was completely unconstrained, which makes it impossible to meaningfully compare the results of the two experiments. In fact, after adjusting the Zhang et al. water chemistry conditions for the expected drift, the isotopic results are not all that different (as discussed in our reply).

4) Again, I agree that using a NBS calibrated pH electrode to calculate carbonate chemistry speciation is far from ideal and there can be a substantial bias, but this would simply shift the data of Zhang et al. shown in Figure 1 horizontally, while it would need vertical shifting (unless the $p\text{CO}_2$ would have been underestimated in an extreme fashion - see below).

Yes, but as we discuss in our reply, our data (PIC and POC) trend downward with increasing $p\text{CO}_2$. Thus, shifting the Zhang et al data horizontally to the right (as would occur when correcting for liquid junction correction of the electrode), these data would fall within the span of our data (see Fig 1 of Zhang et al).

5) As already pointed out in comment #4, potentiometric pH measurements with a NBS-calibrated electrode is not best practice, but can not explain the general isotopic offset between the various data sets. However, for the one data point created by Zhang et al., there is the possibility that $p\text{CO}_2$ was underestimated as of non-ideal NBS-based potentiometric pH measurements (although the magnitude estimated by Liu et al. is most likely extreme). If indeed the case, this would suggest that *Ochrosphaera* could be different to all other coccolithophores tested before. Given the potential problems with the instant ocean seawater and carbonate chemistry calculations mentioned above, and the fact that the data set by Liu et al. appears to stick out rather than the data point generated by Zhang et al., I would rather tend to go with the simplest explanation for the general discrepancy shown in Figure 1, i.e. that isotopic equilibrium was not reached in the original Liu et al. data set.

However, a final verdict would require additional data.

Again, because our data (PIC and POC) trend downward with increasing $p\text{CO}_2$, shifting the Zhang et al data horizontally to the right (as would occur when correcting for liquid junction correction of the electrode) would place the Zhang et al data within the span of our data (see Fig 1 of Zhang et al).

We agree that a final verdict would require additional data, the generation of which we feel is beyond the scope of this Matters Arising - Reply exchange.

We respectfully thank the reviewers for their thoughtful comments and for taking the time to review this exchange.

Referees' comments:

Reviewer #2 (Remarks to the Author):

The authors have replied to all my comments, but appear to have chosen to ignore the possibility pointed out by Zhang et al., that the trend in $\delta^{13}\text{C}_{\text{PIC}}$ they observed with increasing CO_2 could very much be artificially generated by changes in cell density and different degrees of heavy ^{13}C enrichment generated by photosynthesis. This is a crucial point, as the authors' argument that both data sets could tell the same story if Zhang et al. underestimated their CO_2 concentration, critically hinges on the maybe artificially created $\delta^{13}\text{C}_{\text{PIC}}$ trend. But I leave this for the reader to decide.

One final thing, though, the authors should remove their claim that bubbling is the preferred OA manipulation method and stated in the Guide for best practices. It is simply not true. What is preferred is a DIC enrichment, which can be done via CO_2 bubbling or combined $\text{HCO}_3^-/\text{CO}_3^{2-}$ and HCl addition, like in Zhang et al. Both approaches achieve the exact same manipulation in terms of changes to DIC, TA, pH, CO_2

Reviewer #2 (Remarks to the Author):

The authors have replied to all my comments, but appear to have chosen to ignore the possibility pointed out by Zhang et al., that the trend in $\delta^{13}\text{C}_{\text{PIC}}$ they observed with increasing CO_2 could very much be artificially generated by changes in cell density and different degrees of heavy ^{13}C enrichment generated by photosynthesis. This is a crucial point, as the authors' argument that both data sets could tell the same story if Zhang et al. underestimated their CO_2 concentration, critically hinges on the maybe artificially created $\delta^{13}\text{C}_{\text{PIC}}$ trend. But I leave this for the reader to decide.

In this revision, we further clarified this issue in the fourth section of our reply. Unlike Zhang et al.'s experimental setting, we continuously bubbled CO_2 into the system. In this case, similar to what has been proposed by ¹, Rayleigh fractionation of $\delta^{13}\text{C}$ due to photosynthesis may be diluted as the $\delta^{13}\text{C}$ of DIC was continuously replenished in our system and therefore the DIC pool was larger than the initial size confined in the tank.

Additionally, the photosynthesis-driven Rayleigh fractionation might be significant when the coccolithophore density was high in the close system; however, our cell densities actually were low in the treatments. To avoid confusion, we clarified that the '100,000 cell/mL' reported was as a conservative upper bound rather than the coccolithophore concentration measured in this revision.

One final thing, though, the authors should remove their claim that bubbling is the preferred OA manipulation method and stated in the Guide for best practices. It is simply not true. What is preferred is a DIC enrichment, which can be done via CO_2 bubbling or combined $\text{HCO}_3^-/\text{CO}_3^{2-}$ and HCl addition, like in Zhang et al. Both approaches achieve the exact same manipulation in terms of changes to DIC, TA, pH, CO_2

We removed the statement and acknowledged that carbonate-HCl addition method is also one of the OA manipulation methods recommended in the Guide for best practice. Here we want to address the point that "continuous bubbling to the system" instead of a "close system" in Zhang et al. makes a difference in stabilizing the carbonate chemistry, as well as the evolution of carbon isotope composition in the water media.

Reference

- 1 Hermoso, M., Chan, I. Z. X., McClelland, H. L. O., Heureux, A. M. C. & Rickaby, R. E. M. Vanishing coccolith vital effects with alleviated carbon limitation. *Biogeosciences* **13**, 301-312, doi:10.5194/bg-13-301-2016 (2016).